# A new preferentially outcrossing monoicous species of *Volvox* sect. *Volvox* (Chlorophyta) from Thailand

**Hisayoshi Nozaki**[1]*, **Wuttipong Mahakham**[2], **Wirawan Heman**[3], **Ryo Matsuzaki**[4,5], **Masanobu Kawachi**[4]

**1** Department of Biological Sciences, Graduate School of Science, The University of Tokyo, Bunkyo-ku, Tokyo, Japan, **2** Department of Biology, Faculty of Science, Khon Kaen University, Khon Kaen, Thailand, **3** Department of Science and Mathematics, Faculty of Science and Health Technology, Kalasin University, Thailand, **4** Center for Environmental Biology and Ecosystem Studies, National Institute for Environmental Studies, Tsukuba, Ibaraki, Japan, **5** Faculty of Life and Environmental Sciences, University of Tsukuba, Tsukuba, Ibaraki, Japan

* nozaki@bs.s.u-tokyo.ac.jp

**Data Availability Statement:** New sequence data, alignments used for our phylogenetic analyses, and new strains are available under the DDBJ/ENA/GenBank accession numbers (LC546055–

## Abstract

*Volvox* sect. *Volvox* is an interesting group of green algae; it comprises mostly monoicous species, but evidence suggests an evolution towards dioicy. Based on cultured strains originating from Thailand, we describe *Volvox longispiniferus*, a novel species in *Volvox* sect. *Volvox*. This species is distinguished from others in the section by the large number of sperm packets in its monoicous sexual spheroids and by the long spines on its zygote wall. Phylogenetic analyses indicate that *V. longispiniferus* is distinct from the other species of two monophyletic groups within *Volvox* sect. *Volvox*. In addition, the novel species produces more zygotes when different cultures are combined compared with a single culture, suggesting a preference for outcrossing.

## Introduction

Sex is recognized in various eukaryotic lineages and contributes to the mixing of genomes between two individuals, normally designated as male or female based on the production of sperm or eggs, respectively. In some invertebrates, flowering plants, fungi, and algae, both sperm and eggs are produced by the same individual; this is known as "hermaphroditism", "monoecism", "monoicy", or "homothallism" [1–3]. Monoecism in flowering plants has been of interest to evolutionary biologists since the time of Charles Darwin, as self-fertilization rapidly leads to inbreeding depression [4].

Within the green algal genus *Volvox*, *Volvox* sect. *Volvox* exhibits interesting sexual features, with seven monoicous and three dioicous species [5–7], and ancestral state reconstruction suggests its evolution from monoicy toward dioicy [3]. Recently, Hanschen et al. [8] studied a dioicous species of *Volvox* sect. *Volvox* (*V. perglobator*) and further explored the evolution of dioicy in this section.

LC546069), TreeBASE ID (S26186), and NIES Collection strain designations (NIES-4432–NIES-4436), respectively. All other relevant data are within the paper and its Supporting Information files.

**Funding:** This study was supported by a Grants-in-Aid for Scientific Research (grant numbers 16H02518 for HN and 19K22446 for HN) from the Ministry of Education, Culture, Sports, Science and Technology (MEXT)/Japan Society for the Promotion of Science (JSPS) KAKENHI (https://www.jsps.go.jp/english/e-grants/), and the Applied Taxonomic Research Center (grant number: ATRC KKUR6309; http://atrc.sc.kku.ac.th/), Department of Biology, Faculty of Science, Khon Kaen University. The funders had no role in study design, data collection and analysis, decision to publish, or preparation of the manuscript.

**Competing interests:** The authors have declared that no competing interests exist.

During field surveys in Thailand, WM collected water samples from which strains of *Volvox* sect. *Volvox* were cultured. Under experimental culture conditions, we induced the production of monoicous sexual spheroids, which in turn produce zygotes. Intriguingly, this species exhibited an apparent preference for outcrossing. Based on morphological characteristics and molecular phylogeny, the strains were identified as a novel species, *Volvox longispiniferus* Nozaki & Mahakham sp. nov. Here we describe the morphology, phylogeny, taxonomy, and outcrossing preference of this species.

## Materials and methods

### Ethics statement

WM collected colonial volvocine green algae from the water column in a marsh in Thailand. Collection was undertaken in accordance with the Plant Variety Protection Act, B.E. 2542 (1999), Section 53, Department of Agriculture, Thailand, which addresses the collection of plants for research, study, or experimentation for non-commercial purposes. Analysis of Thai materials was conducted in accordance with a Memorandum of Understanding between the University of Tokyo and Khon Kaen University for international cooperative research on the systematics, phylogenetics, and evolution of freshwater green algae in Thailand (2017–2021). A Material Transfer Agreement was arranged between Khon Kaen University and the University of Tokyo, with WM as the provider scientist and HN as the recipient scientist.

### Establishment of cultures and morphological observations

Water samples (pH 6.85; temperature 30.5˚C) were collected from a marsh in Nong Ya Ma, Yang Talat District, Kalasin Province, Thailand (16˚ 28′ 14.55′′ N, 103˚ 16′ 25.55′′ E), on 1 November 2019 (Table 1). Clonal cultures of *V. longispiniferus* (strains 1101-NZ-3, 1101-NZ-4, 1101-NZ-5, 1101-NZ-14, and 1101-RM-5) were established from the water sample using the pipette washing method [9]. Cultures were grown in $18 \times 150$ mm screw-cap tubes containing 10–11 mL artificial freshwater-6 (AF-6) or *Volvox* thiamin acetate (VTAC) medium [10] at 25˚C on a 14:10 hour light: dark (L:D) schedule, under cool-white fluorescent lamps (with color temperature of 5000 K) at an intensity of 80–130 µmol $m^{-2}$ $s^{-1}$. They were at first maintained in AF-6 medium. For observing morphological details, possible bacterial contamination was removed from the cultures by picking up a young spheroid still within the parental spheroid and washing the young spheroid several times with fresh medium using a micropipette; the young spheroid was then grown in 10–11 mL VTAC medium. Asexual spheroids were observed in actively growing cultures in VTAC medium, as described previously [11]. To induce production of sexual spheroids, 0.3–0.8 mL actively growing culture in VTAC medium (the volume depending upon the number and density of the spheroids in the culture) was inoculated with 10–11 mL urea soil *Volvox* thiamine (USVT) medium [12]. This culture was grown at 20˚C on a 14:10 hour L:D schedule under cool-white fluorescent lamps at an intensity of 50–70 µmol $m^{-2}$ $s^{-1}$. Sexual spheroids typically developed within 5 days. To enhance formation and maturation of zygotes, immature sexual spheroids cultured under the aforementioned conditions were transferred to fresh USVT medium and cultured at 25˚C on a 14:10 hour L:D schedule. For measurements of rates of zygote formation (fertilization rates) in eggs per spheroid, three types of cultures of sexual spheroids ("sexual cultures 1–3"; see below) were prepared and grown at 25˚C on a 14:10 hour L:D schedule.

Sexual culture 1: 1.0 mL cultures including immature sexual spheroids from two different cultures were mixed and inoculated into 10–11 mL fresh USVT medium in Petri dishes ($57 \times 16$ mm) (60mm/Non-treated Dish, IWAKI AGC TECHNO GLASS, Shizuoka, Japan).

**Table 1. List of species/strains used in the present phylogenetic analyses (Figs 4–6).**

| Species | Sample/strain designation | Origin of sample/strain | DDBJ/ENA/GenBank Accession number | | |
|---|---|---|---|---|---|
| | | | ITS-1, 5.8S rDNA and ITS-2 | *rbcL* | *psbC* |
| *Volvox longispiniferus* sp. nov. from Thailand | 1101-NZ-3[a] (= NIES-4432) | Water sample collected from a marsh in Nong Ya Ma, Yang Talat District, Kalasin Province, Thailand (water temperature 30.5˚C; pH 6.85; 16˚ 28´ 14.55´´ N, 103˚ 16´ 25.55´´ E) in 1 November 2019. | LC546055[b] | LC546060[a] | LC546065[b] |
| | 1101-NZ-4[a] (= NIES-4433) | | LC546056[b] | LC546061[b] | LC546066[b] |
| | 1101-NZ-5[a] (= NIES-4434) | | LC546057[b] | LC546062[b] | LC546067[b] |
| | 1101-NZ-14[a] (= NIES-4435) | | LC546058[b] | LC546063[b] | LC546068[b] |
| | 1101-RM-5[a] (= NIES-4436) | | LC546059[b] | LC546064[b] | LC546069[b] |
| *Volvox barberi* | UTEX 804 | USA | AB663341 | D86835 | AB044477 |
| *Volvox capensis* | NIES-3874 | USA | LC034074 | LC033870 | LC033872 |
| *Volvox kirkiorum* | NIES-2740 | Japan | AB663324 | AB663322 | AB663323 |
| *Volvox ferrisii* | NIES-2736 | Japan | AB663336 | AB663334 | AB663335 |
| *Volvox globator* | SAG 199.80 (= UTEX 955) | USA | AB663340 | D86836 | AB044478 |
| *Volvox perglobator* | TucsonVspTf | USA | MG429137 | KY489662 | KY489659 |
| *Volvox rousseletii* from South Africa | UTEX 1862 (= NIES-734) | South Africa | AB663342 | D63448 | AB044479 |
| *Volvox rousseletii* from Japan | NIES-4336 | Japan | LC493797 | LC493808 | LC493810 |
| *Volvox* sp. Sagami | NIES-4021 | Japan | LC191308 | LC191316 | LC191326 |
| *Colemanosphaera angeleri* | NIES-3382 | Japan | | AB905592 | AB905598 |
| *Colemanosphaera charkowiensis* | NIES-3383 | Japan | | AB905591 | AB905598 |
| *Platydorina caudata* | NIES-728 (= UTEX 1658) | USA | | D86828 | AB044494 |

[a] Established in this study.

[b] Sequenced in this study.

Sexual culture 2: 2.0 mL culture including immature sexual spheroids from a single culture was inoculated into 10–11 mL fresh USVT medium in Petri dishes (57 × 16 mm).

Sexual culture 3: a single immature sexual spheroid was isolated by a micropipette and inoculated into 0.5 mL fresh USVT medium in a tissue culture plate (MULTIWELL™ 48 well, Becton Dickinson, NJ, USA).

After 6–8 days, formation of zygotes was examined under the light microscope. For statistical analyses, the Kolmogorov-Smirnov test of normality was performed for the three data sets "sexual cultures 1–3" by Social Science Statistics https://www.socscistatistics.com/tests/kolmogorov/default.aspx (S1 Table). Student's *t*-test of Social Science Statistics was subjected to pairs of the data sets because they did not differ significantly from that which is normally distributed (S1 Table).

Light microscopy was conducted using the BX60 microscope (Olympus, Tokyo, Japan) equipped with Nomarski interference optics. Spheroid cells were counted as described previously [5,13].

## Molecular experiments

To infer the phylogenetic position of *V. longispiniferus* within *Volvox* sect. *Volvox*, we evaluated the internal transcribed spacer (ITS) regions of nuclear ribosomal DNA ((rDNA); ITS-1, 5.8S rDNA, and ITS-2) and two chloroplast genes (the large subunit of Rubisco (*rbcL*) and photosystem II CP43 apoprotein (*psbC*) genes) as described previously [7]. Sequences were

determined based on direct sequencing of polymerase chain reaction (PCR) products from a disrupted cell solution, using KOD One PCR Master Mix (Toyobo, Osaka, Japan). The PCR parameters for *rbcL* and *psbC* amplification were 2 minutes at 94˚C, followed by 45 cycles of 10 seconds at 98˚C, 30 seconds at 50˚C, and 30 seconds at 68˚C; those for rDNA ITS amplification were 2 minutes at 94˚C, followed by 40 cycles of 10 seconds at 98˚C, 30 seconds at 66˚C, and 30 seconds at 68˚C.

To assess the phylogeny of rDNA ITS regions, chloroplast genes (*rbcL* plus *psbC*), and a combined dataset from rDNA ITS regions and the two chloroplast genes, we analyzed the operational taxonomic units of the species, samples, and strains listed in Table 1. Sequences were aligned as described previously [6,12,14], and the alignments are available in TreeBASE (www.treebase.org/treebase-web/home.html; Study ID S26186). We followed previously described methods [6] for outgroup designation in the *rbcL-psbC* phylogeny and used the tree topology from this phylogeny to designate the root of *Volvox* sect. *Volvox* in the analyses of rDNA ITS regions and the combined dataset. Maximum-likelihood (ML) analyses based on sequence alignments from all three datasets were performed using MEGA X [15] with best-fitted models (K2+G for rDNA ITS, GTR+G for *rbcL-psbC*, and T92+G for the combined data set) selected by MEGA X and 1000 bootstraps [16]. In addition, Bayesian inference (BI) of the alignments was determined using MrBayes 3.2.6 [17], as described previously [14]. A partitioned analysis was implemented in BI of *rbcL-psbC* and the combined data set. Models for BI were SYM+G for rDNA ITS, and F81, JC, and GTR+I+G for first, second, and third codon positions of *rbcL-psbC*, respectively, selected by hierarchical likelihood ratio test using MrModeltest 2.3 [18].

The secondary structures of ITS-2 were predicted as described previously [6,12,14,19].

## Nomenclature

The electronic version of this article in Portable Document Format (PDF) in a work with an ISSN or ISBN constitutes a published work according to the International Code of Nomenclature for algae, fungi, and plants (Article 29.1) [20]; hence, the new names contained in the electronic publication of a PLOS ONE article are effectively published under that Code from the electronic edition alone, so there is no longer any need to provide printed copies.

## Results

### Asexual spheroids

Mature asexual spheroids of *V*. *longispiniferus* in the present cultures were subspherical or ovoid in shape, with 5600–8200 somatic cells embedded in individual sheaths at the periphery of the gelatinous matrix (Fig 1A–1C). Spheroids were up to 830 μm long. Somatic cells had two flagella and a cup-shaped chloroplast with a single basal pyrenoid and a single eyespot up to 10 μm long and were connected by cytoplasmic bridges (Fig 1C and 1D). Cytoplasmic bridges were thicker than the flagella (Fig 1C). The anterior somatic cells of the spheroids were laterally subspherical or trapezoidal, and the cell length was shorter than or nearly equal to the cell width (Fig 1D). In general, 7–11 gonidia, or developing embryos, were present in the posterior two-thirds of asexual spheroids (Fig 1A and 1B). Gonidia of the next generation were not evident in the developing embryo either during or immediately after inversion (Fig 1E).

### Sexual reproduction

Induced sexual spheroids were monoicous, forming both sperm packets and eggs (Fig 2A–2C). Mature sexual spheroids were subspherical or ovoid in shape and up to 800 μm long, with

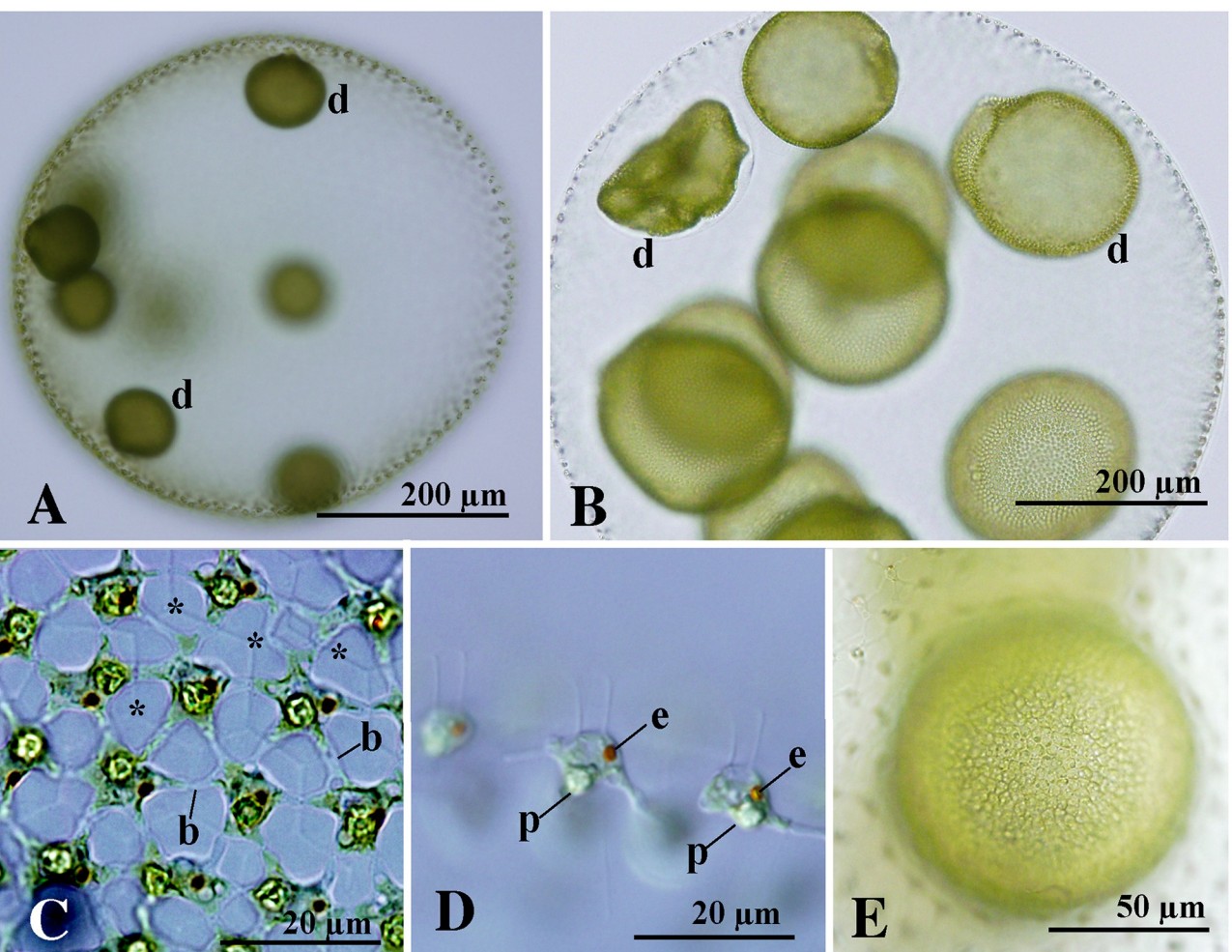

**Fig 1. Asexual spheroids of *Volvox longispiniferus* sp. nov. strain 1101-NZ-5 from Thailand.** (A, B, E) Bright-field microscopy. (C, D) Nomarski differential interference contrast microscopy. (A) Spheroid with developing embryos (d). (B) Fully matured spheroid with inverting daughter spheroids (d). (C–E) Part of spheroids. (C) Front view of somatic cells with thick cytoplasmic bridges (b) and individual sheaths (asterisks). (D) Side view of anterior somatic cells showing eyespot (e) and pyrenoid (p) in the chloroplast. (E) Surface view of compact embryo just after inversion. Note that differentiation of gonidia of the next generation is not evident.

4900–7800 somatic cells arranged at the periphery. Spheroids had 25–58 eggs and 5–16 sperm packets distributed randomly within the posterior four-fifths of the spheroid. Differences in proportion of eggs and sperm packets in a sexual spheroid were not seen between isolates of *V. longispiniferus*. Sperm packets often swam outside the spheroid and attached to the surface of the sexual spheroid; packets then gradually dissociated to become individual sperm that penetrated the spheroid, potentially for fertilization. Irrespective of the presence or absence of fertilization, eggs matured nearly simultaneously within a single sexual spheroid (Fig 2D). Mature zygotes were reddish–brown in color with a spiny cell wall (Fig 2E), but some mature eggs did not secret a cell wall possibly due to the lack of fertilization. Fully developed spines were 12–14 μm long and were straight or slightly curved, with an acute apex (Fig 2E). Zygotes were 40–48 μm in diameter, excluding spines. Hatching of the zygotes was not observed.

Under the sexual induction conditions used here, abundant zygotes were formed when the two different cultures including immature sexual spheroids were mixed and grown in USVT medium at 25°C. More than 50% of eggs in sexual spheroids developed into matured zygotes

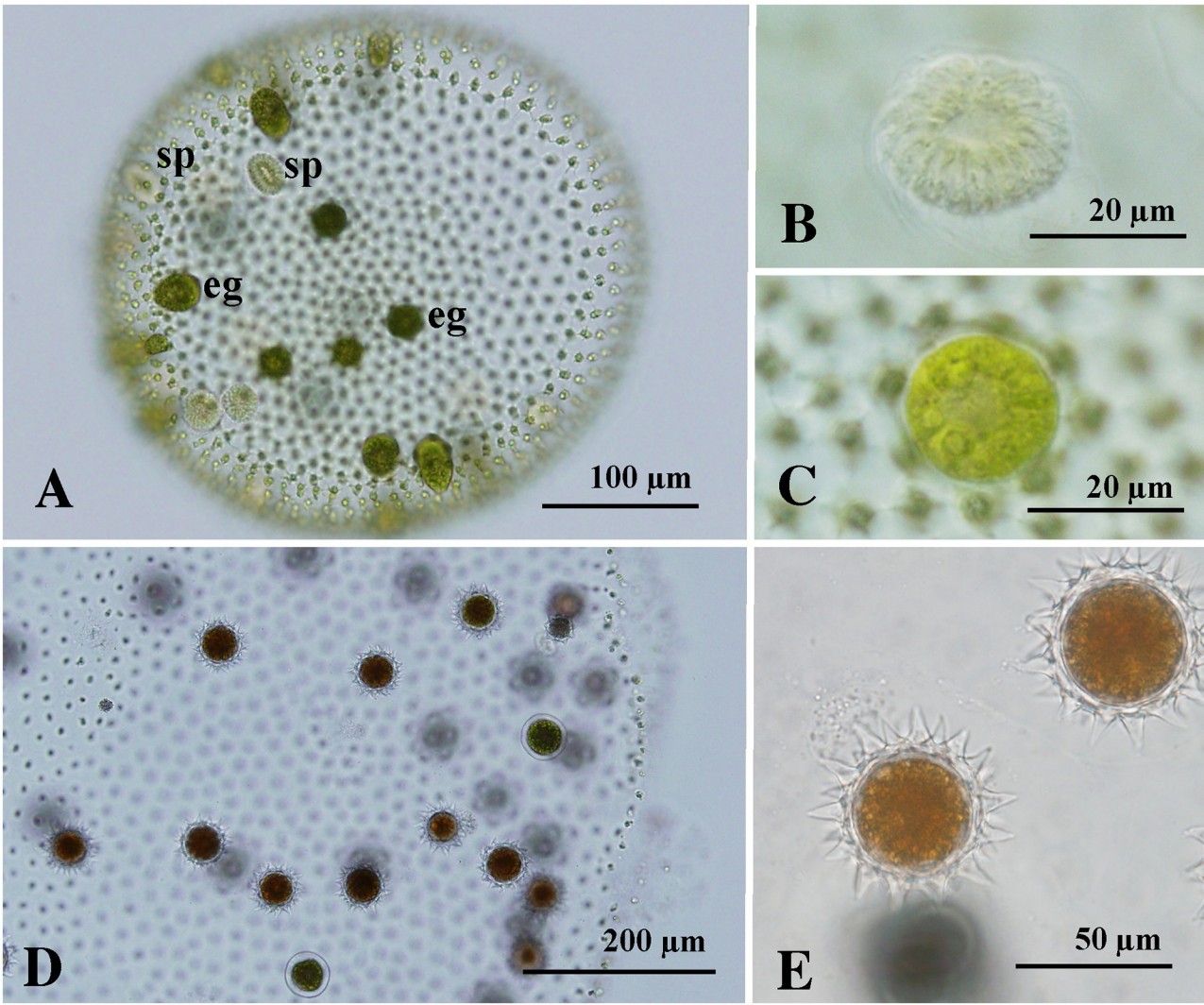

**Fig 2. Sexual reproduction of *Volvox longispiniferus* sp. nov. from Thailand.** (A, B, D, E) Bright-field microscopy. (C) Nomarski differential interference contrast microscopy. (A) Young monoicous spheroid showing eggs (eg) and sperm packets (sp). Strain 1101-NZ-4. (B) Sperm packet in monoicous spheroid. Strain 1101-NZ-4. (C) Egg in monoicous spheroid. Strain 1101-NZ-4. (D) Sexual spheroid with matured zygotes. Strains 1101-NZ-4 x 1101-NZ-5. (E) Matured zygotes with long, acute spines on zygote walls. Strains 1101-NZ-4 x 1101-NZ-5.

in such mixed culture when the strains were newly established (three months after the establishment; Figs 2D and 3A–3C). However, ratios of zygote formation in sexual spheroids decreased during the maintenance of cultures (four months after the establishment).

Comparative measurements of ratios of zygote formation in eggs per sexual spheroid were carried out after four months from the strain establishment. At that time, 6–25% (average 17%) of eggs in a spheroid developed into potential zygotes (walled cells) when two different clonal cultures were mixed and grown in USVT medium for zygote formation and maturity (sexual culture 1) (S1 Table). By contrast, 0–19% (average 12%) of eggs in a spheroid developed into potential zygotes within a single sexually induced culture (sexual culture 2) (Fig 3D). When a single immature sexual spheroid was isolated and inoculated into fresh USVT medium (sexual culture 3), 3–19% (average 11%) of eggs developed into potential zygotes (Fig 3E and 3F). Based on the zygote formation rates in eggs per sexual spheroid from these three

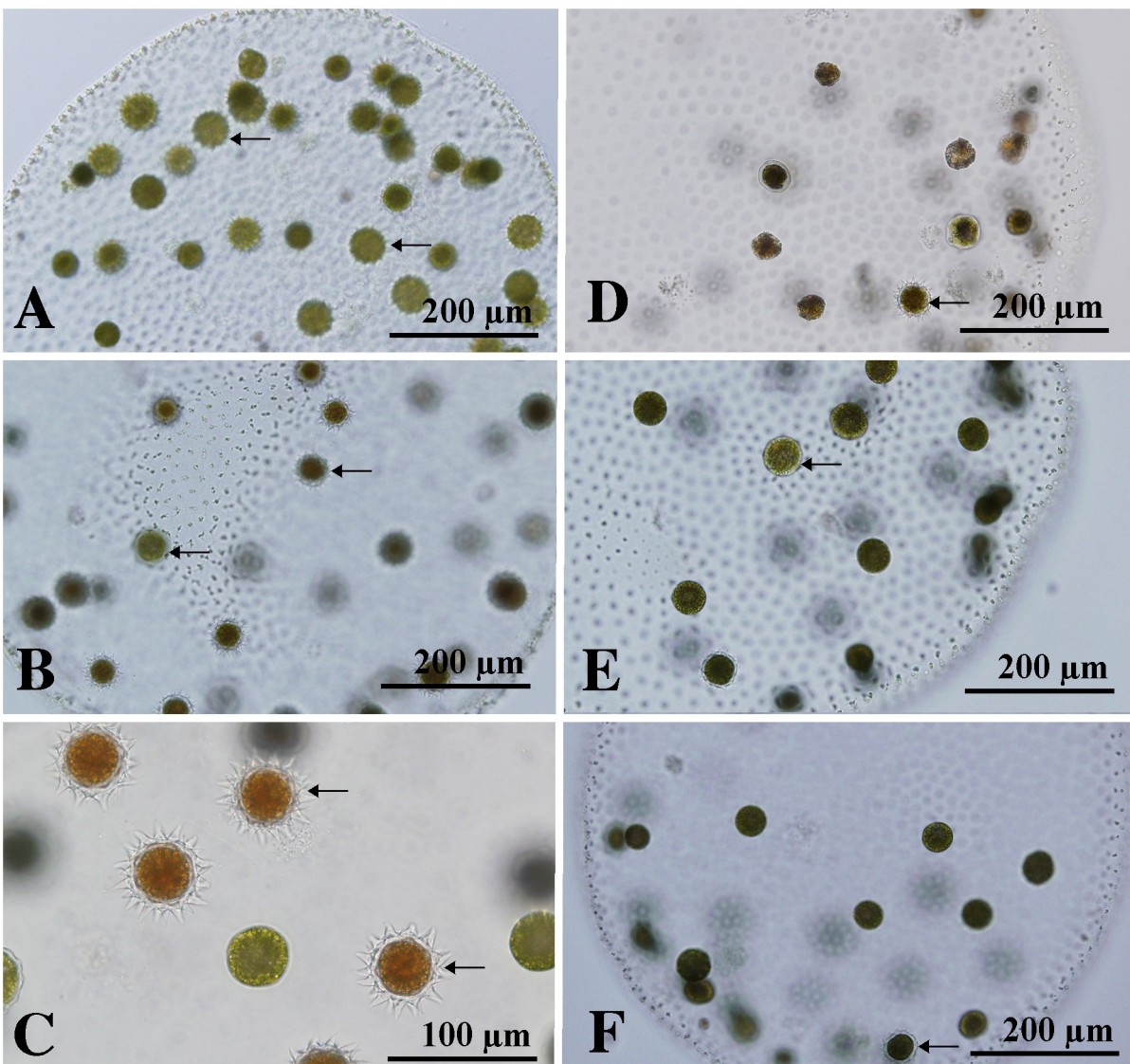

**Fig 3. Comparison of development of potential zygotes (walled cells, arrows) in sexual spheroids of *Volvox longispiniferus* sp. nov. between mixture of two different cultures (A–C) and a single clonal culture (D–F).** (A–C) Strains 1101-NZ-4 x 1101-NZ-5. Note more than half of eggs developing into zygotes. (A) Four days after mixture of sexually induced cultures. (B) Six days after the mixture. (C) Eight days after the mixture. (D) Eight-day-old sexual spheroid in a single culture (strain 1101-NZ-5). (E, F) Six-day-old, singly isolated sexual spheroids of strain 1101-NZ-4 (E) and strain 1101-NZ-5 (F). Note less formation of zygotes in sexual spheroids that were isolated singly.

types of sexual cultures, statistical t-tests were carried out. Significant difference [corrected probability values (based on Bonferroni correction) < 0.05] in the rate was detected between sexual cultures 1 and 3 (S1 Table).

## Molecular phylogeny

All five strains of *V. longispiniferus* exhibited identical *psbC* and *rbcL* sequences; however, the nuclear rDNA ITS region of the five strains comprised two distinct types that differed with respect to three nucleotides in the ITS-1 region. The phylogenetic positions of

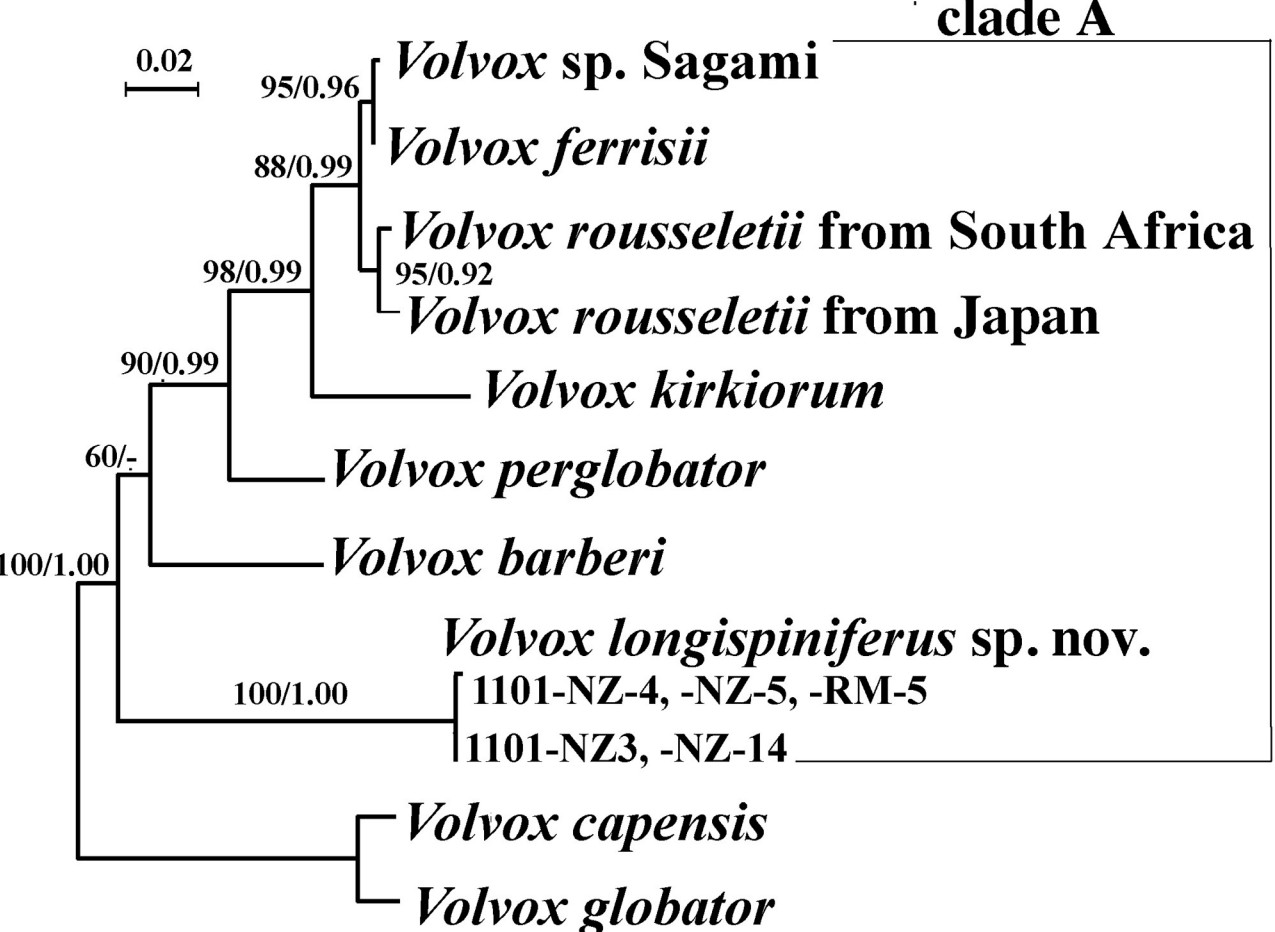

**Fig 4. Phylogenetic analysis of *Volvox longispiniferus* sp. nov. from Thailand, based on Maximum Likelihood (ML) method of the Internal Transcribed Spacer (ITS) regions of nuclear ribosomal DNA (rDNA) (ITS-1, 5.8S rDNA, and ITS-2).** Branch lengths are proportional to the evolutionary distances that are indicated by the scale bar. Numbers in left and right sides at branches represent bootstrap values (50% or more) based on 1000 replications of ML and posterior probabilities (0.90 or more) by Bayesian inference, respectively.

*V. longispiniferus* based on the ITS region and *rbcL-psbC* gene sequences are shown in Figs 4 and 5, respectively. Phylogenetic relationships within *Volvox* sect. *Volvox* were essentially identical between the two trees, and two robust sister clades were resolved, one clade composed of *V. globator* and *V. capensis* and the other (clade A) containing *V. rousseletii*, *V. perglobator*, *V.* sp. Sagami, *V. ferrisii*, *V. kirkiorum*, *V. barberi*, and *V. longispiniferus*. Within clade A, a large monophyletic group composed of *V. rousseletii*, *V. perglobator*, *V.* sp. Sagami, *V. ferrisii*, and *V. kirkiorum* was resolved, with 68–90% bootstrap values in ML and 0.99–1.00 posterior probability in BI. However, the phylogenetic positions of *V. longispiniferus* and *V. barberi* were not robustly resolved within clade A. The combined dataset demonstrated moderate support for the sister relationship between *V. longispiniferus* and others within clade A, with a 73% bootstrap value in ML and 0.92 posterior probability in BI (Fig 6).

When the secondary structures of ITS-2 were compared, two compensatory base changes were recognized between *V. longispiniferus* and *V. barberi* and three between *V. longispiniferus* and *V. perglobator* (S1 and S2 Figs).

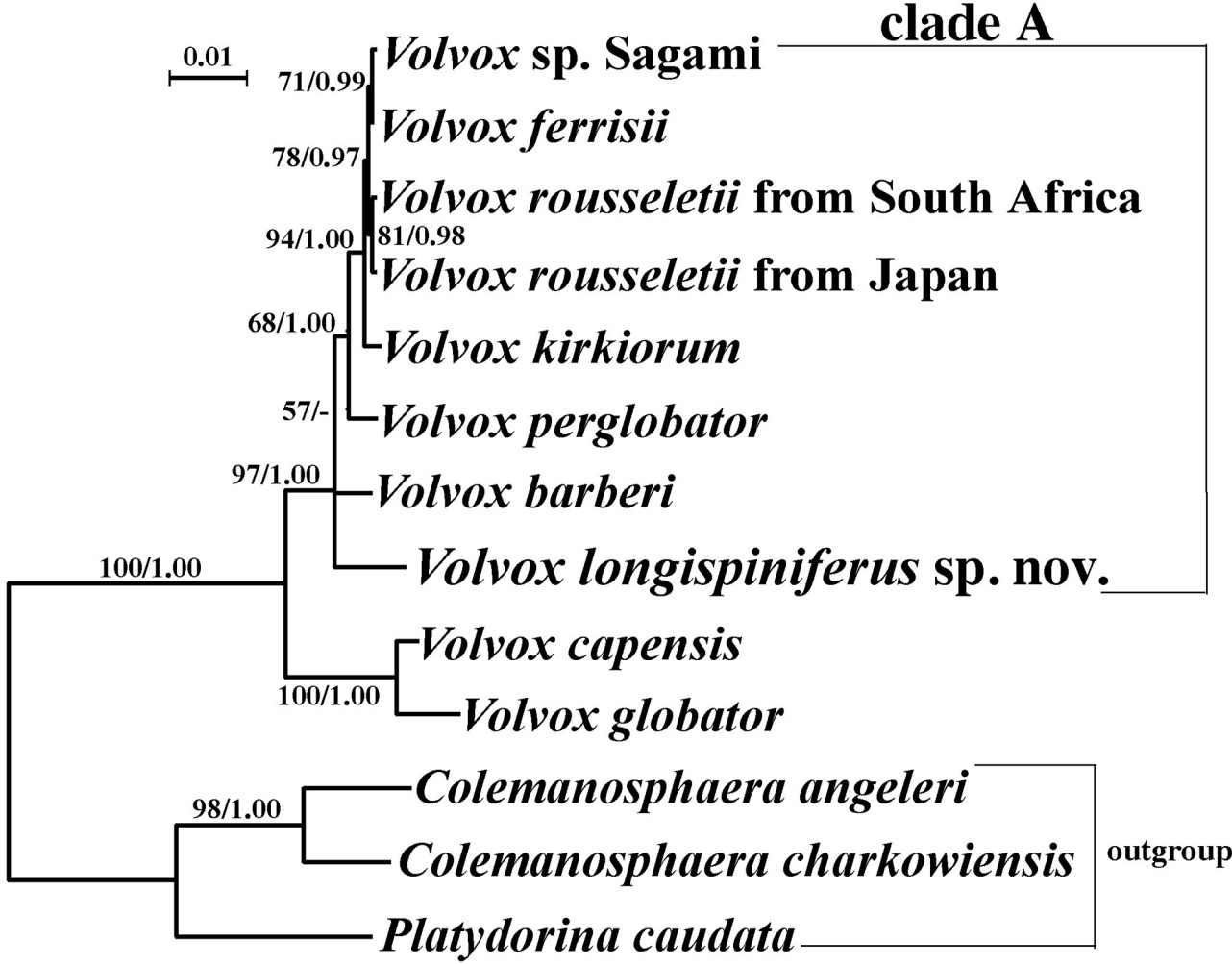

**Fig 5. Phylogenetic analysis of *Volvox longispiniferus* sp. nov. from Thailand, based on Maximum Likelihood (ML) method of two chloroplast genes (*rbcL* and *psbC*).** Branch lengths are proportional to the evolutionary distances that are indicated by the scale bar. Numbers in left and right sides at branches represent bootstrap values (50% or more) based on 1000 replications of ML and posterior probabilities (0.90 or more) by Bayesian inference, respectively.

## Discussion

Seven monoicous species and one monoicous morphological type had previously been recognized in *Volvox* sect. *Volvox* (S2 Table) [6]. Among these, *V. longispiniferus* is morphologically similar to *V. merrillii* by virtue of its long zygote spines (> 10 μm) and the production of asexual spheroids with more than 10 gonidia or daughter spheroids (S2 Table). However, *V. longispiniferus* differs from *V. merrillii* with respect to the number of eggs and sperm packets in monoicous sexual spheroids. *Volvox merrillii* typically produces sexual spheroids with 60 or more eggs and up to eight sperm packets (S2 Table). By comparison, the sexual spheroids of *V. longispiniferus* have 25–58 eggs and 5–16 sperm packets. In addition, sizes of zygotes in *V. longispiniferus* are larger than that of *V. merrillii* (S2 Table). Zygotes of *V. longispiniferus* measure 40–48 μm in diameter (without spines) whereas those of *V. merrillii* are 37–40 μm in diameter (S2 Table). When numbers of eggs in sexual spheroids are focused (S2 Table), *V. longispiniferus* (with 25–58 eggs) is similar to *V. kirkiorum* (with 20–80 eggs). However, *V. kirkioru*m has short zygote spines (< 10 μm) and pear-shaped to ovoid anterior somatic cells

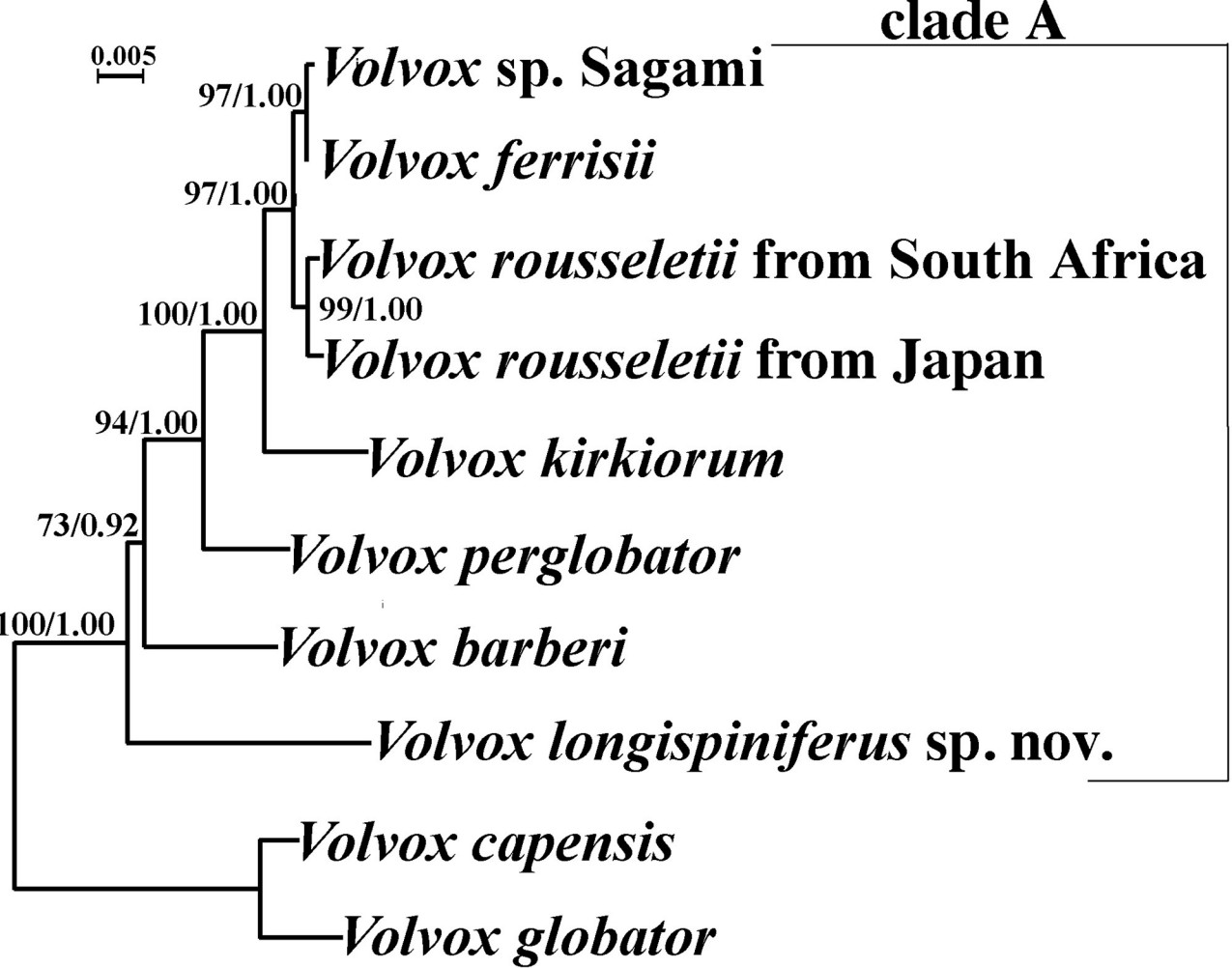

**Fig 6. Phylogenetic analysis of *Volvox longispiniferus* sp. nov. (strains 1101-NZ-4, 1101-NZ-5 and 1101-RM-5) from Thailand, based on Maximum Likelihood (ML) method of combined data set from the Internal Transcribed Spacer (ITS) regions of nuclear ribosomal DNA (rDNA) (ITS-1, 5.8S rDNA, and ITS-2) and two chloroplast genes (*rbcL* and *psbC*).** Branch lengths are proportional to the evolutionary distances that are indicated by the scale bar. Numbers in left and right sides at branches represent bootstrap values (50% or more) based on 1000 replications of ML and posterior probabilities (0.90 or more) by Bayesian inference, respectively.

(S2 Table). Anterior somatic cells of *V. longispiniferus* are subspherical or trapezoidal in shape (Fig 1D). Therefore, *V. longispiniferus* is clearly distinguished from other monoicous species in *Volvox* sect. *Volvox* by the long (12–14 μm) acute spines on the zygote wall and the production of fewer than 60 eggs in monoicous spheroids (S2 Table). Although *V. longispiniferus* was examined in only clonal cultured materials, no clonal cultures have been established and molecular data are lacking for *V. merrillii* (S2 Table). *V. amboensis* also lacks clonal cultural studies and molecular data (S2 Table). But this species is morphologically distinct from other monoicous species in *Volvox* sect. *Volvox* in having a large sexual spheroid measuring up to 2000 μm long with usually more than 200 eggs (S2 Table). Thus, eight morphological species and one morphological type with monoicous sexual spheroids are now recognized in *Volvox* sect. *Volvox* (S2 Table). However, further cultural studies and molecular data of *V. merrillii* and *V. amboensis* are needed to resolve their detailed taxonomic and phylogenetic positions within *Volvox* sect. *Volvox*.

Our phylogenetic analyses demonstrate that *V. longispiniferus* is weakly to moderately separated from other species within *Volvox* sect. *Volvox* (Figs 4–6). Strictly speaking, the phylogenetic relationship between *V. longispiniferus* and *V. barberi* is not well resolved, but both species belong to the robust clade A in this section (Figs 4–6). Furthermore, *V. barberi* and *V. perglobator* represent two major lineages within the clade A excluding *V. longispiniferus* (Fig 6). Comparison of the secondary structures of ITS-2 rDNA demonstrated two or three compensatory base changes between *V. longispiniferus* and *V. barberi* or between *V. longispiniferus* and *V. perglobator*, respectively (S2 Fig). These values suggest enough genetic differentiation to recognize *V. longispiniferus* as different species within clade A. Thus, *V. longispiniferus* comprises a new, morphologically and phylogenetically distinct species within *Volvox* sect. *Volvox*.

Monoicous sexual spheroids in *V. capensis* from the United States have 2–6 sperm packets and 70–100 eggs (S2 Table). Although sperm packets in this species do not escape from the sexual spheroids, nearly all eggs in the sexual spheroids develop into spiny-walled zygotes [12]. A similar phenomenon has been observed in *V. ferrisii*, another monoicous species that has 3–5 sperm packets and 100–150 eggs in the monoicous spheroids (S2 Table) [14]. By comparison, the ratio of sperm packets to eggs in *V. longispiniferus* is relatively high (5–16 sperm packets and 25–56 eggs; S2 Table); in addition, *V. longispiniferus* exhibits produces more zygotes when different cultures are combined compared with a single isolated spheroid (Fig 3; S1 Table). Therefore, *V. longispiniferus* may preferentially outcross via fertilization by sperm from other sexual spheroids. Preferential outcrossing also occurs in flowering plants [21], in which effective fertilization in outcrossing populations of monoicous plants relies on substantial production and transfer of pollen or sperm to other individuals. Conversely, self-fertilizing individuals do not require the same abundant production of pollen or sperm because of the higher probability of fertilization within a single individual [21]. Thus, the preferential outcrossing exhibited by *V. longispiniferus* may be indicative of the initial stages of transition to an outcrossing monoicous species.

## Taxonomic treatment

### *Volvox longispiniferus* Nozaki & Mahakham sp. nov.

Asexual spheroids subspherical or ovoid in shape, containing 5600–8200 somatic cells embedded in individual sheaths of the gelatinous matrix, with seven to 11 gonidia or developing embryo in the posterior two-thirds of the spheroid, measuring up to 830 μm long. Cells connected by thick cytoplasmic bridges. Anterior somatic cells of the spheroid were laterally subspherical or trapezoidal. Gonidia of the next generation not evident in the developing embryo either during or immediately after inversion. Sexual spheroids monoicous, 4900–7800-celled, with 25–58 eggs and 5–16 sperm packets distributed randomly within the posterior four-fifths of the spheroid, measuring up to 830 μm long. Eggs maturing nearly simultaneously within a single sexual spheroid. Mature zygotes with a spiny cell wall, measuring 40–48 μm in diameter (excluding spines). Fully developed spines 12–14 μm long, straight or slightly curved with an acute apex.

Holotype: Fig 2A, depicting a monoicous spheroid with eggs and sperm packets of strain 1101-NZ-4. This strain is available as NIES-4433 from the Microbial Culture Collection at the National Institute for Environmental Studies, Japan [10]. The holotype of a name of this new species is an effectively published illustration since it is impossible to preserve a specimen that would show the features, especially the monoicous sexual spheroid, attributed to *V. longispiniferus* (Article 40.5 of International Code of Nomenclature for algae, fungi, and plants [20]).

Strains examined: 1101-NZ-3 (= NIES-4432), 1101-NZ-4 (= NIES-4433), 1101-NZ-5 (= NIES-4434), 1101-NZ-14 (= NIES-4435) and 1101-RM-5 (= NIES-4436) (Table 1).

Etymology: The species epithet "*longispiniferus*" meaning "with long spines" for zygote wall morphology.

Type locality: A marsh in Nong Ya Ma, Yang Talat District, Kalasin Province, Thailand (16˚ 28′ 14.55′′ N, 103˚ 16′ 25.55′′ E). Water samples were collected by WM on 1 November 2019.

## Supporting information

**S1 Fig. The secondary structure of nuclear ribosomal DNA (rDNA) Internal Transcribed Spacer 2 (ITS-2) transcript of five strains of *Volvox longispiniferus* sp. nov., including the 3' end of the 5.8S ribosomal RNA (rRNA) and the 5' end of the large Subunit of rRNA (LSU rRNA).**
(DOCX)

**S2 Fig. Comparison of helices of the secondary structure of nuclear ribosomal DNA internal transcribed spacer 2 transcripts between *Volvox longispiniferus* sp. nov. and its related strain/species (Figs 4–6).**
(DOCX)

**S1 Table. Results of zygote formation in eggs per sexual spheroid from three types of sexually induced cultures of *Volvox longispiniferus* sp. nov.**
(DOCX)

**S2 Table. Comparison of *Volvox longispiniferus* and previously described monoicous morphological type and species of *Volvox* sect. *Volvox*.**
(DOCX)

## Acknowledgments

We would like to thank Dr Noppawan Nounjan, Department of Biology, Faculty of Science, Khon Kaen University for her assistance in field collection.

## Author Contributions

**Data curation:** Hisayoshi Nozaki, Ryo Matsuzaki.

**Formal analysis:** Hisayoshi Nozaki, Ryo Matsuzaki.

**Funding acquisition:** Wuttipong Mahakham, Masanobu Kawachi.

**Investigation:** Hisayoshi Nozaki, Wuttipong Mahakham, Wirawan Heman, Ryo Matsuzaki.

**Methodology:** Hisayoshi Nozaki.

**Project administration:** Hisayoshi Nozaki, Wuttipong Mahakham.

**Resources:** Wuttipong Mahakham, Masanobu Kawachi.

**Supervision:** Hisayoshi Nozaki, Wuttipong Mahakham, Wirawan Heman, Masanobu Kawachi.

**Validation:** Hisayoshi Nozaki.

**Visualization:** Hisayoshi Nozaki, Ryo Matsuzaki.

**Writing – original draft:** Hisayoshi Nozaki, Wuttipong Mahakham.

**Writing – review & editing:** Hisayoshi Nozaki, Wuttipong Mahakham, Wirawan Heman, Ryo Matsuzaki, Masanobu Kawachi.

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
