## [Decision Letter · Decision Letter 0]

4 Jun 2020

PONE-D-20-14342

A new preferentially outcrossing monoecious species of Volvox sect. Volvox (Chlorophyta) from Thailand

PLOS ONE

Dear Dr. Nozaki,

Thank you for submitting your manuscript to PLOS ONE. After careful consideration, we feel that it has merit but does not fully meet PLOS ONE’s publication criteria as it currently stands. Therefore, we invite you to submit a revised version of the manuscript that addresses the points raised during the review process.

I and two experts have read and reviewed your manuscript.  All of us are in agreement that it is suitable for publication pending minor revision and adequate responses to the Reviewers' comments. If you intend to include data on out-crossing please make sure to describe the experiments adequately in the Methods and pay attention to replication and statistics needed to support your conclusions.

I have two additional minor comments.

1. Please indicate in your Methods whether the cultures were maintained axenically. If not, could the presence of microbial contaminants have affected the outcomes of the crossing experiments?

2. Terminology in the field is changing.  Although traditionally monoecy/monecious dioecy/dioecious have been used for volvocine algae, these could be replaced by monoicy/monoicous and dioicy/dioicous that are used for plants that have haploid-phase sex determination (e.g. bryophytes); while the former terms are reserved for plant species with diploid-phase sex determination. 10.1016/j.tplants.2018.06.005. It is fine if you leave your terminology as is, but wanted to bring this alternative nomenclature system to your attention.

We look forward to receiving your revised manuscript.

Kind regards,

James G. Umen, Ph. D.

Academic Editor

PLOS ONE

Journal Requirements:

'W.M. gratefully acknowledge financial support from Department of Biology & Applied Taxonomic Research Center, Faculty of Science, Khon Kaen University, Thailand.'

'This study was supported by a Grants-in-Aid for Scientific Research (grant numbers 16H02518 for HN and 19K22446 for HN) from the Ministry of Education, Culture, Sports, Science and Technology (MEXT)/Japan Society for the Promotion of Science (JSPS) KAKENHI (https://www.jsps.go.jp/english/e-grants/). The funders had no role in study design, data collection and analysis, decision to publish, or preparation of the manuscript'

Reviewers' comments:

Reviewer's Responses to Questions

**Comments to the Author**

1. Is the manuscript technically sound, and do the data support the conclusions?

Reviewer #1: Yes

Reviewer #2: Yes

2. Has the statistical analysis been performed appropriately and rigorously? 

Reviewer #1: No

Reviewer #2: Yes

3. Have the authors made all data underlying the findings in their manuscript fully available?

Reviewer #1: Yes

Reviewer #2: Yes

4. Is the manuscript presented in an intelligible fashion and written in standard English?

Reviewer #1: Yes

Reviewer #2: Yes

5. Review Comments to the Author

Reviewer #1: Review of Nozaki et al; A new preferentially outcrossing monoecious species of Volvox sect. Volvox (Chlorophyta) from Thailand.

This manuscript describes a new monoecious, or hermaphroditic, species of green algae in the Volvox section Volvox of the volvocine green algae, Volvox longispiniferus. The authors provide clear and detailed methods and results regarding both asexual and sexual lifecycles of V. longispiniferus, and provide very interesting results on the preferential outcrossing of this hermaphrodite. This is good research and forms a workhorse species description that is worth publication upon a few improvements. My only major comment is that the preferential outcrossing suggestion is not sufficiently discussed and should be expanded upon.

This experiment is very interesting, as well as the suggestion that V. longispiniferus may be on a transition towards heterothallic obligate outcrossing. For the suggestion that V. longispiniferus may be transitioning to heterthallic obligate outcrossing, please clarify whether or not differences in proportion of eggs and sperm are seen between isolates of V. longispiniferus.

However, these results, despite making up the title and much of the discussion are in the supplemental file, not the main document. The authors should move Figure S1 into the main text. Furthermore, they may have enough quantitative data regarding egg number and fertilization rates to perform a statistical test, likely t test or Mann-Whitney U test depending on whether the data is normal or not, on the differences between fertilization rates between single colony and mixed colony experiments. Lastly, for these experiments, please add to the methods section clarifying whether a similar number of colonies and density of colonies could be expected based on adding 0.3-0.8 mL of culture to the experiment.

Minor comments:

1. Hanschen et al 2018 (see below) also described a Volvox section Volvox species and further explored the evolution of dioecy in Volvox section Volvox.

2. Did the authors observe hatching of the fertilized or unfertilized zygotes?

3. Please provide (brief) justification that this species is not V amboensis or V. kirkiorum in the main text.

4. Is it possible to add known geographical distribution to Table S1?

5. Page 5, line 94, please clarify which model of evolution was used in phylogenetic analyses and if a partitioning scheme was implemented in the psbC/rbcL phylogeny.

6. Figure 1, panel D, please ensure consistency of labelling eyespot (figure) or stigma (figure legend).

Reviewer #2: This is a nice, brief paper describing a new species of Volvox with some interesting features. It is well-written, understandable, and rigorous. I have just a few concerns, all but one of which are trivial.

My one substantive concern is regarding the justification for considering the new strains a separate species from V. merrillii. The phylogenetic position of V. merrillii is unclear and genetic data are unavailable (understandably, since — I think— this species is not available for examination), so we have to make do without those lines of evidence. The only justification that is given is the number of eggs and sperm packets in sexual colonies. Table S1 indicates a few other differences, for example number of cells in asexual spheroids, shape of anterior cells, and number of cells in sexual spheroids, but these are not discussed in the main text. The reason these differences don’t convince me that V. longispiniferus and V. merrillii are distinct species is that these traits are known to vary in response to culture conditions and by the length of time the strain has been cultured. The descriptions of V. merrillii are based on two very old (but authoritative) papers, and it’s not clear whether the culture conditions and the age of the strains are similar to the present study. Dr. Nozaki is, by far, the world’s leading expert on the phylogeny and taxonomy of Volvox and its relatives, and if he says they are distinct species they probably are, but the current version of the manuscript does not sufficiently explain the justification. I realize that the unfortunate lack of V. merrillii cultures limits the information that can be obtained. I would nevertheless like to see a much expanded discussion of the differences between V. longispiniferus and V. merrillii that addresses intraspecific variation, variation in response to strain age and culture conditions, and any other factors that might inform the distinctness of the putative new species.

Minor comments:

Regarding V. merrillii, a short explanation of the reason genetic and more recent morphological data are not available would be great.

The preference for outcrossing figures prominently in the abstract, but it is barely mentioned in the main text, with no methods provided and the relevant figure relegated to a supplement. Since this is the aspect of the paper that is most likely to interest the poor benighted souls who study organisms less interesting than Volvox, I urge the authors to consider discussing the experiment in more detail and moving the figure to the main text.

Line 19: I would like to know what Charles Darwin said about inbreeding depression, but Correspondence Volume 9, at 645 pages, is a bit much to wade through in search of the passage the authors have in mind. I know page numbers are generally only provided for direct quotes, but I hope the editors will allow an exception.

23: water samples IN which strains were cultured, or water samples FROM which strains were cultured?

53: please provide the color temperature in K of the lamps.

Table 1: I love that the authors have included the NIES strain designations. This is a service to future Volvox researchers.

110: it’s not clear to me whether these numbers are based on collected or cultured spheroids. The world’s leading authority on the phylogeny and taxonomy of Volvox and its relatives once told me that spheroids in culture rarely obtain the sizes of their wild-caught sisters.

139-140: What happened to the unfertilized eggs?

223: If the branching order in Fig. 3 is correct, V. longispiniferus is equally related to all of the other species in Clade A. There is no reason to focus this comparison on V. barberi in particular, since V. ferrisii (for example) is just as closely related to V. longispiniferus.

224: Some discussion of the value of compensatory base changes in ITS-2 in distinguishing species is needed.

6. PLOS authors have the option to publish the peer review history of their article (what does this mean?). If published, this will include your full peer review and any attached files.

Reviewer #1: No

Reviewer #2: No

---

## [Author Response · Author response to Decision Letter 0]

11 Jun 2020

PONE-D-20-14342

A new preferentially outcrossing monoecious species of Volvox sect. Volvox (Chlorophyta) from Thailand

PLOS ONE

Dear Dr. Nozaki,

Thank you for submitting your manuscript to PLOS ONE. After careful consideration, we feel that it has merit but does not fully meet PLOS ONE’s publication criteria as it currently stands. Therefore, we invite you to submit a revised version of the manuscript that addresses the points raised during the review process.

I and two experts have read and reviewed your manuscript. All of us are in agreement that it is suitable for publication pending minor revision and adequate responses to the Reviewers' comments. If you intend to include data on out-crossing please make sure to describe the experiments adequately in the Methods and pay attention to replication and statistics needed to support your conclusions.

Response: The data have been statistically examined and detailed methods for outcrossing have been described in the revised manuscript.

I have two additional minor comments.

1. Please indicate in your Methods whether the cultures were maintained axenically. If not, could the presence of microbial contaminants have affected the outcomes of the crossing experiments?

Response: We cannot say that the cultures were maintained axenically because of lack of examination of cultures grown in bacterial media. However, the cultures used for crossing experiments (S1 Table of the revised manuscript) did not show apparent bacteria when grown in acetate-containing medium VTAC. Thus, there is little possibility that microbial contaminants have affected the outcomes of the crossing experiments. For clarifying the removal of bacteria from the original cultures, details of the culture methods have been described in the Method section of the revised manuscript.

2. Terminology in the field is changing. Although traditionally monoecy/monecious dioecy/dioecious have been used for volvocine algae, these could be replaced by monoicy/monoicous and dioicy/dioicous that are used for plants that have haploid-phase sex determination (e.g. bryophytes); while the former terms are reserved for plant species with diploid-phase sex determination. 10.1016/j.tplants.2018.06.005. It is fine if you leave your terminology as is, but wanted to bring this alternative nomenclature system to your attention.

Response: The terminology has been revised as suggested.

We look forward to receiving your revised manuscript.

Kind regards,

James G. Umen, Ph. D.

Academic Editor

PLOS ONE

Journal Requirements:

Response: Done as suggested.

'W.M. gratefully acknowledge financial support from Department of Biology & Applied Taxonomic Research Center, Faculty of Science, Khon Kaen University, Thailand.'

'This study was supported by a Grants-in-Aid for Scientific Research (grant numbers 16H02518 for HN and 19K22446 for HN) from the Ministry of Education, Culture, Sports, Science and Technology (MEXT)/Japan Society for the Promotion of Science (JSPS) KAKENHI (https://www.jsps.go.jp/english/e-grants/). The funders had no role in study design, data collection and analysis, decision to publish, or preparation of the manuscript'

Response: We have removed any funding-related text from the manuscript. We would like to update your Funding Statement. Currently, your Funding Statement reads as follows: 

'This study was supported by a Grants-in-Aid for Scientific Research (grant numbers 16H02518 for HN and 19K22446 for HN) from the Ministry of Education, Culture, Sports, Science and Technology (MEXT)/Japan Society for the Promotion of Science (JSPS) KAKENHI (https://www.jsps.go.jp/english/e-grants/), and the Applied Taxonomic Research Center (grant number: ATRC KKUR6309; http://atrc.sc.kku.ac.th/), Department of Biology, Faculty of Science, Khon Kaen University. The funders had no role in study design, data collection and analysis, decision to publish, or preparation of the manuscript'

Response: Done as suggested.

Reviewers' comments:

Reviewer's Responses to Questions

Comments to the Author

1. Is the manuscript technically sound, and do the data support the conclusions?

Reviewer #1: Yes

Reviewer #2: Yes

2. Has the statistical analysis been performed appropriately and rigorously? 

Reviewer #1: No

Reviewer #2: Yes

Response: For outcross experiments, statistical analyses by Student’s t-test based on the Kolmogorov-Smirnov test of normality have been carried out in the revised manuscript.

3. Have the authors made all data underlying the findings in their manuscript fully available?

Reviewer #1: Yes

Reviewer #2: Yes

4. Is the manuscript presented in an intelligible fashion and written in standard English?

Reviewer #1: Yes

Reviewer #2: Yes

5. Review Comments to the Author

Reviewer #1: Review of Nozaki et al; A new preferentially outcrossing monoecious species of Volvox sect. Volvox (Chlorophyta) from Thailand.

This manuscript describes a new monoecious, or hermaphroditic, species of green algae in the Volvox section Volvox of the volvocine green algae, Volvox longispiniferus. The authors provide clear and detailed methods and results regarding both asexual and sexual lifecycles of V. longispiniferus, and provide very interesting results on the preferential outcrossing of this hermaphrodite. This is good research and forms a workhorse species description that is worth publication upon a few improvements. My only major comment is that the preferential outcrossing suggestion is not sufficiently discussed and should be expanded upon.

This experiment is very interesting, as well as the suggestion that V. longispiniferus may be on a transition towards heterothallic obligate outcrossing. For the suggestion that V. longispiniferus may be transitioning to heterthallic obligate outcrossing, please clarify whether or not differences in proportion of eggs and sperm are seen between isolates of V. longispiniferus.

Response: Differences in proportion of eggs and sperm packets in a sexual spheroid were not seen between isolates of V. longispiniferus. This situation has been described in the revised manuscript.

However, these results, despite making up the title and much of the discussion are in the supplemental file, not the main document. The authors should move Figure S1 into the main text. Furthermore, they may have enough quantitative data regarding egg number and fertilization rates to perform a statistical test, likely t test or Mann-Whitney U test depending on whether the data is normal or not, on the differences between fertilization rates between single colony and mixed colony experiments. 

Response: In the revised version, Figure S1 has been moved into the main text (Figure 3), and statistical tests have been performed to detect differences in fertilization rates between single spheroid and mixed culture experiments, as well as between the unmixed and mixed culture experiments. Based on the Kolmogorov-Smirnov test of normality, all of the three data sets of fertilization rates per spheroid (mixed culture, unmixed culture and single spheroid experiments) do not differ significantly from those which is normally distributed. Thus, Student’s t-tests have been carried out to detect significant difference (p < 0.05) in fertilization rates per spheroid between single colony and mixed culture experiments, as well as between the unmixed and mixed culture experiments. These experimental procedures and results have been described in the main text and S1 Table newly prepared in the revised manuscript.

Lastly, for these experiments, please add to the methods section clarifying whether a similar number of colonies and density of colonies could be expected based on adding 0.3-0.8 mL of culture to the experiment.

Response: Cultures of Volvox are very unstable in growing conditions principally due to the pre-culture. Thus, a similar number of spheroids and density of spheroids could be expected based on adding 0.3-0.8 mL of culture to the experiment by using a loupe or naked eye. The Method section has been revised to clarify this point.

Minor comments:

1. Hanschen et al 2018 (see below) also described a Volvox section Volvox species and further explored the evolution of dioecy in Volvox section Volvox.

Response: The following paper has been cited in the Introduction section in the revised manuscript.

Hanschen ER, Davison DR, Ferris　PJ, Michod RE. On the rediscovery of Volvox perglobator (Volvocales, Chlorophyceae) and the evolution of outcrossing from self-fertilization. Evol Ecol Res. 2018; 19: 299–318.

2. Did the authors observe hatching of the fertilized or unfertilized zygotes?

Response: No. This situation has been described in the revised manuscript.

3. Please provide (brief) justification that this species is not V amboensis or V. kirkiorum in the main text.

Response: Discussions regarding justification that V. longispiniferus is not V amboensis or V. kirkiorum have been described in the Discussion section of the revised manuscript. 　

4. Is it possible to add known geographical distribution to Table S1?

Response: Geographical distribution has been added to S2 Table in the revised manuscript.

5. Page 5, line 94, please clarify which model of evolution was used in phylogenetic analyses and if a partitioning scheme was implemented in the psbC/rbcL phylogeny.

Response: Details of ML analyses and BI, especially models used, have been described in the Method section of the revised manuscript. 

6. Figure 1, panel D, please ensure consistency of labelling eyespot (figure) or stigma (figure legend).

Response: Stigma (s) has been changed eyespot (e) throughout the text.

Reviewer #2: This is a nice, brief paper describing a new species of Volvox with some interesting features. It is well-written, understandable, and rigorous. I have just a few concerns, all but one of which are trivial.

My one substantive concern is regarding the justification for considering the new strains a separate species from V. merrillii. The phylogenetic position of V. merrillii is unclear and genetic data are unavailable (understandably, since — I think— this species is not available for examination), so we have to make do without those lines of evidence. The only justification that is given is the number of eggs and sperm packets in sexual colonies. Table S1 indicates a few other differences, for example number of cells in asexual spheroids, shape of anterior cells, and number of cells in sexual spheroids, but these are not discussed in the main text. The reason these differences don’t convince me that V. longispiniferus and V. merrillii are distinct species is that these traits are known to vary in response to culture conditions and by the length of time the strain has been cultured. The descriptions of V. merrillii are based on two very old (but authoritative) papers, and it’s not clear whether the culture conditions and the age of the strains are similar to the present study. Dr. Nozaki is, by far, the world’s leading expert on the phylogeny and taxonomy of Volvox and its relatives, and if he says they are distinct species they probably are, but the current version of the manuscript does not sufficiently explain the justification. I realize that the unfortunate lack of V. merrillii cultures limits the information that can be obtained. I would nevertheless like to see a much expanded discussion of the differences between V. longispiniferus and V. merrillii that addresses intraspecific variation, variation in response to strain age and culture conditions, and any other factors that might inform the distinctness of the putative new species.

Response: We are very happy to know that the reviewer realizes that the unfortunate lack of V. merrillii cultures limits the information that can be obtained. Based on the comment, much expanded discussion of the differences between V. longispiniferus and V. merrillii has been described in the Discussion section of the revised manuscript. However, there have been no clonal culture studies of Volvox merrillii, we cannot address intraspecific variation, variation in response to strain age and culture conditions.

Minor comments:

Regarding V. merrillii, a short explanation of the reason genetic and more recent morphological data are not available would be great.

Response: Explanation of the reason genetic and more recent morphological data are not available has been described in the Discussion section of the revised manuscript.

The preference for outcrossing figures prominently in the abstract, but it is barely mentioned in the main text, with no methods provided and the relevant figure relegated to a supplement. Since this is the aspect of the paper that is most likely to interest the poor benighted souls who study organisms less interesting than Volvox, I urge the authors to consider discussing the experiment in more detail and moving the figure to the main text.

Response: In the revised version, Figure S1 has been moved into the main text (Figure 3), and statistical tests have been performed to detect differences in fertilization rates between single spheroid and mixed culture experiments, as well as between the unmixed and mixed culture experiments. Our Student’s t-tests have detected significant difference (p < 0.05) in fertilization rates per spheroid between single colony and mixed culture experiments, as well as between the unmixed and mixed culture experiments. These procedures and results have been described in the main text and S1 Table newly prepared in the revised manuscript.

Line 19: I would like to know what Charles Darwin said about inbreeding depression, but Correspondence Volume 9, at 645 pages, is a bit much to wade through in search of the passage the authors have in mind. I know page numbers are generally only provided for direct quotes, but I hope the editors will allow an exception.

Response: The citation is based on the PLoS ONE Journal style. Thus, revision is not needed.

23: water samples IN which strains were cultured, or water samples FROM which strains were cultured?

Response: Revised as suggested.

53: please provide the color temperature in K of the lamps.

Response: the color temperature in K (5000 K) has been described in the revised manuscript.

Table 1: I love that the authors have included the NIES strain designations. This is a service to future Volvox researchers.

Response: Thank you!

110: it’s not clear to me whether these numbers are based on collected or cultured spheroids. The world’s leading authority on the phylogeny and taxonomy of Volvox and its relatives once told me that spheroids in culture rarely obtain the sizes of their wild-caught sisters.

Response: In the revised manuscript, we have clarified that the numbers are based on cultured spheroids.

139-140: What happened to the unfertilized eggs?

Response: Possible unfertilized eggs have been described in the revised manuscript.

223: If the branching order in Fig. 3 is correct, V. longispiniferus is equally related to all of the other species in Clade A. There is no reason to focus this comparison on V. barberi in particular, since V. ferrisii (for example) is just as closely related to V. longispiniferus.

Response: V. barberi and V. perglobator were selected as two phylogenetically separated representatives of clade A (excluding V. longispiniferus). Since we found two or three CBC in ITS-2 secondary structure between V. longispiniferus and V. barberi or between V. longispiniferus and V. perglobator, respectively, we consider that there is no need to compare V. longispiniferus with other species in clade A to discuss the genetic independency of V. longispiniferus within clade A. In the revised manuscript, such discussion has been added in the Discussion section.

224: Some discussion of the value of compensatory base changes in ITS-2 in distinguishing species is needed.

Response: Some discussion of the values of compensatory base changes in ITS-2 has been added to discuss the genetic independency of V. longispiniferus within Volvox sect. Volvox in the revised manuscript.

6. PLOS authors have the option to publish the peer review history of their article (what does this mean?). If published, this will include your full peer review and any attached files.

Do you want your identity to be public for this peer review? For information about this choice, including consent withdrawal, please see our Privacy Policy.

Reviewer #1: No

Reviewer #2: No

---

## [Editor Report · Decision Letter 1]

19 Jun 2020

A new preferentially outcrossing monoicous species of Volvox sect. Volvox (Chlorophyta) from Thailand

PONE-D-20-14342R1

Dear Dr. Nozaki,

We’re pleased to inform you that your manuscript has been judged scientifically suitable for publication and will be formally accepted for publication once it meets all outstanding technical requirements.

Kind regards,

James G. Umen, Ph. D.

Academic Editor

PLOS ONE
---

## [Editor Report · Acceptance letter]

24 Jun 2020

PONE-D-20-14342R1 

A new preferentially outcrossing monoicous species of *Volvox* sect. *Volvox* (Chlorophyta) from Thailand 

Dear Dr. Nozaki:

I'm pleased to inform you that your manuscript has been deemed suitable for publication in PLOS ONE. Congratulations! Your manuscript is now with our production department. 

Kind regards, 

on behalf of

Dr. James G. Umen 

Academic Editor

PLOS ONE